# Daily and Seasonal Activity Patterns of Plateau Pikas (*Ochotona curzoniae*) on the Qinghai–Tibet Plateau, China, and Their Relationship with Weather Condition

**DOI:** 10.3390/ani13101689

**Published:** 2023-05-18

**Authors:** Rui Zhou, Rui Hua, Zhuangsheng Tang, Limin Hua

**Affiliations:** 1Key Laboratory of Grassland Ecosystem of the Ministry of Education, College of Grassland Science, Gansu Agricultural University, Lanzhou 730070, China; 2Qinghai Academy of Animal Science and Veterinary Medicine, Qinghai University, Xining 810016, China

**Keywords:** activity patterns, plateau pika, camera trap, activity level, weather condition

## Abstract

**Simple Summary:**

Animals are subject to the daily and seasonal fluctuations of climatic factors that affect their thermoregulation. The purpose of the present study was to determine the activity patterns of wild plateau pikas (*Ochotona curzoniae*), with a special emphasis on the effects of thermal conditions. We set 14 camera traps in a coniferous forest, Yakushima, and monitored them for a total of 850 camera days between October 2017 and September 2018. During the cold season, pikas concentrated their activity around noon and exhibited unimodal patterns. During the warm season, pikas concentrated their activity in the morning and afternoon and exhibited bimodal patterns. Plateau pikas occupied a cool and windless microclimate during cold seasons and a cool and humid microclimate during warm seasons to avoid excessive cooling or overheating pressure. These results indicate that changes in the microclimatic conditions of their habitats (specifically, increases in temperature) might significantly restrict the daytime activity of plateau pikas. These results can provide baseline information for future research on pikas, especially that which is focused on their behavioral ecology and the effects of climate change in some Chinese alpine regions.

**Abstract:**

Exploring the activity patterns of small mammals is important for understanding the survival strategies of these animals, such as foraging and mating. The purpose of the present study was to determine the activity of free-living plateau pikas (*Ochotona curzoniae*) in different months and seasons (cold and warm seasons), with a particular emphasis on the effects of weather condition. Based on a camera-trapping survey conducted from October 2017 to September 2018, we evaluated the activity patterns and activity levels of plateau pikas inhabiting the eastern Qinghai–Tibet Plateau in China. The effects of environmental factors on the activity of plateau pikas were examined using the generalized additive mixed model (GAMM). The results showed that: (1) The plateau pikas exhibited unimodal patterns of activity during the cold season (October–April). During the warm season (May–September), the activity patterns of the plateau pikas were bimodal. Their activity levels were highest in June. (2) During the cold season, their activity levels rose gradually over the course of the day to a peak near noon, and they were not significantly higher after sunrise than they were before sunset. During the warm season, their activity peaks were in the morning and afternoon, and their activity levels were substantially lower after sunrise than they were before sunset. (3) The plateau pikas were more active under conditions with lower ambient temperatures and precipitation during the cold and warm seasons. While relative air humidity was positively correlated with the activity of the plateau pikas during the warm season, wind speed was negatively correlated with the pikas’ activity during the cold season. Overall, these results collectively indicate that plateau pikas occupy habitats with cool and less windy microclimates during the cold season, and with cool and moist microclimates during the warm season. Information on the time allocation of pikas’ activity levels during different seasons should provide a baseline for understanding their potential for adaptation to climate change.

## 1. Introduction

Animals allocate time to their various activities through daily and annual activity patterns to most efficiently cope with varying energy demands and environmental changes that may threaten their costs and benefits of foraging, movement, and mating [1,2]. To meet so many costs and needs, animals use optimized behavioral strategies to limit unnecessary energy loss; for example, reducing activity and foraging times, seeking thermal shelter, or adjusting the budget and pattern of their diurnal activities are important thermoregulatory behavioral mechanisms to prevent hyperthermia and hypothermia [3,4]. However, while these thermoregulatory strategies can cushion organisms from extreme weather, they may force individuals to make tradeoffs between meeting their resource, calorie, or security requirements [5], which may lead to locally or seasonally adjusted activity patterns [6]. The study of an animal’s activity patterns can reflect its natural physio–biological characteristics and can help to elucidate the environmental adaptations of the animal [7].

Animal activities may be influenced by specific abiotic variables, and climate factors are particularly important because they directly affect the cost of thermoregulation [8,9]. For example, ambient temperatures can affect animals indirectly through their food supply or predator behavior, but they can also directly affect thermoregulation [10]. Endothermic maintain their body temperature by producing heat inside of their body [11]. Heat production is energetically costly, so behavioral thermoregulation, behaviors that minimize the cost of physiological thermoregulation, is also important [12]. These behaviors include choosing a microenvironment that is thermally favorable and inhibiting heat loss by changing one’s posture [13]. Precipitation is another climate factor that contributes to thermoregulatory costs. It can affect the activities of both ectothermic and endothermic animals [14]. Studying both how animals respond to their environment and the plasticity in their behaviors allows us to better understand patterns of species coexistence, resource partitioning, and predator avoidance [15,16,17].

Despite the importance of studying animal activity, the accurate measurement of field activity is technically difficult, resulting in a wide range of approaches. Methods for measuring the activity levels of small mammals have traditionally depended either on direct observation [18] or on the use of laboratory apparatuses such as running wheels [19]. Direct observation allows researchers to obtain detailed information, but it can influence the behavior of subjects and is not possible when animals are in darkness, dense vegetation, or shelter [20]. In addition, live trapping increases sample size, but decreases the activity of individuals sampled and may result in data skewed by hunger or other variables that affect trapping performance [21]. To overcome these challenges, infrared camera trapping (CT) has received increasing attention during the last decade [22]. This survey method has provided a reliable, concealed, and intuitive method of wildlife investigation that greatly reduces the workload of investigation [23]. CT is becoming increasingly popular in ecology and can provide a great amount of information [24,25]. With continuous improvement in CT function and a reduction in equipment costs, CT has gradually become the main method for studying wildlife behavior [26].

Plateau pikas (*Ochotona curzoniae*) are small, diurnal lagomorphs that inhabit alpine meadows on the Qinghai–Tibet Plateau (QTP), China [27]. They have crucial ecological significance for biodiversity maintenance in alpine rangeland ecosystems and for improving soil quality on the QTP [28]. However, this species is regarded as a pest that leads to the degradation of the rangeland ecosystem in the QTP, and it has been largely eliminated by rodenticides [29,30]. Pikas are social mammals that live in groups of adults and their young in a family unit [31]. Within a population, these families consist of a variable number of adult breeding males and females, and individual families may exhibit monogamous, polygynous, or polyandrous mating systems [32]. Like all lagomorphs, plateau pikas do not hibernate during the long montane winter; instead, they survive on stores of haypiles that they stockpile from late July to September [33]. Pikas have high metabolic rates, which is a valuable feature for preserving and maintaining body temperatures during harsh winters [34]. Studying the activity patterns of plateau pikas is of great significance for researchers to better understand the survival strategies of this species under changes to its own and the external environment. There are several important problems in measuring the activity of plateau pikas. Their small sizes and the confines of their tight burrow tunnel limit the sizes and shapes of devices they can carry. Indeed, we are not aware of any field studies on the levels and patterns of activity in plateau pikas that have measured activity throughout the day and night. The only measures of activity have been conducted in a laboratory [35,36]. Studies on the plateau pika’s activity pattern and its influencing factors under field conditions are still lacking.

In this study, we set up 14 infrared cameras in an alpine meadow on the eastern edge of the QTP from October 2017 to September 2018 to monitor the activity patterns of wild pikas. The purpose of our study was to determine the activity patterns of wild plateau pikas at various time scales, and we placed special emphasis on the role of climatic factors (e.g., ambient temperature, relative air humidity, wind speed, and precipitation). We examined seasonal variations, daily variations, and the immediate effects of precipitation. We examined the hypothesis that pikas modify their activity in response to thermal stress [9]. We predict that: (1) during the cold season, pikas shift their activity from the cooler mornings to the warmer afternoons, and their activity is concentrated around the warmest time of day (i.e., noon), (2) during the warm season, pikas decrease their activity at noon, and their activity is concentrated around the cooler time of day (i.e., the morning and afternoon), (3) plateau pikas decrease their activity when the temperature and precipitation are high. 

## 2. Materials and Methods

### 2.1. Study Area

This study was conducted in the eastern section of the northern QTP in the Gansu Province of China from October 2017 to September 2018. A study site (N 33°50′23″, E 102°08′48″) with an area of 4 ha was selected for field sampling (Figure 1). The altitude of the study area is 3434 m a.s.l. Most precipitation (average annual precipitation: 643.9 mm) occurred in July, August, and September [37] (Figure 2A). The air temperature ranged from −16.5 °C to 17.2 °C during the cold season (October to April, average temperature = 0.68 ± 6.87 °C) and from 2.1 °C to 20.2 °C during the warm season (May to September, average temperature = 11.83 ± 3.91 °C). The temperature peaked at 16:00 and was lowest at 7:00 (Figure 2B). The dominant vegetation consisted of typical QTP, alpine meadow vegetation, including a variety of sedges (e.g., *Kobresia pygmaea* C. B. Clarke) and forbs (e.g., *Potentilla anserina* L. var. *anserina*), and a lower percentage of shrubs (*Potentilla fruticosa* Linn. var. *arbuscula*). Forbs and graminoids were widely available and utilized by the pikas [37]. The average population density of the plateau pikas in the study area was 265 burrows per ha, which is a moderate density for Maqu County (Zhou et al., 2023 [38]).

### 2.2. Climatic Data Record

Climatic factors were recorded with an automatic data logger (Kestrel Co., Ltd., Cottleville, MO, USA) within the study site. The data logger recorded ambient temperature, ambient relative humidity, and wind speed at 1 h intervals every day during the study period (October 2017–September 2018). The annual precipitation and daylength during the study period were downloaded from the website of the China Meteorological Administration (http://data.cma.cn/ (accessed on 11 November 2018)).

### 2.3. Camera Trapping

We installed 14 camera traps (East Red Eagle, Shenzhen Ereagle Technology Co., Ltd, Shenzhen, China), each targeting the entrance to a burrow belonging to a single pika family. The cameras were operated for 5 consecutive days every month between October 2017 and September 2018. The cameras were placed at a distance of 1 m from the observation point and 0.5 m above the ground. The lens and the ground met at a top-view angle of more than 15°. The erection direction was southeast or northwest, the camera’s field of view covered a specific area (camera detection area is 13.08 m^2^) around the observation point, and there were no large shrubs or herbaceous plants. The infrared camera settings included 24-h photo mode, the date and time, and an automatic continuous shooting frequency of 5 s. A specially designed spreadsheet was used to record the data from each camera, which included the number of photos taken, the length of photo interval, and the subject being photographed.

### 2.4. Data Analyses

For each camera-trap recording, we identified the species, date, time and camera station. Pikas in pictures from the same camera were considered to be independent when they were taken either at least 30 min apart or at shorter intervals but unambiguously concerning different individuals [39]. The sampling effort totaled 840 trap days and 14 camera trap sites. A total of 14 camera traps were originally set, and two malfunctioned in June and October. We repaired these cameras and used them in subsequent monitoring. The seasons were defined as a warm season and cold season. 

We used the ‘overlap’ R package [40] (and, thus, the method developed by Ridout and Linkie [26]) to determine the daily activity patterns of plateau pikas during the cold and warm seasons then to quantify the overlap between these two patterns. Accordingly, the 24 h density curve of each season was estimated using a von Mises kernel (default smoothing parameter = 1) [41]. Subsequently, the area (Δ) lying under both density curves was assessed using the estimator Δ4. This area ranges from 0 (no overlap) to 1 (complete overlap) [26,39]. Its 95% confidence interval (CI) was further computed using a smoothed bootstrap with 10,000 resamples [40].

Moreover, we divided the 24 h cycle into six periods: one hour before sunrise, one hour after sunrise, midday, one hour before sunset, one hour after sunset, and midnight [42]. Sunrise and sunset times were obtained from the getSunlightTimes function of ‘suncalc’ R package [43]. Activity levels (the proportion of time that pikas spent active) were then estimated for each period using the fitact function of the ‘activity’ R package and compared between periods using the Wald test performed with the compareAct function of the same package [20]. The same method was used to compare the activity levels of the plateau pikas between months, seasons, and time periods.

To examine abiotic factors affecting the activity of the plateau pikas, we used the gamm function in the mgcv package of R, 3.0.1, to conduct generalized additive mixed models (GAMMs) [44], family = poisson, link function = log. We used the number of independent detections per day for each camera as a response variable, and used air temperature (TEM), wind speed (WS), relative air humidity (RH), precipitation (PRE), and day length (DL) as fixed factors. To control for the possible effects of spatial variation, we added the camera ID as a random factor. Twenty-three candidate models were envisaged for the cold season (October–April), and twenty for the warm season (May–September) (Appendix A). Models based on combinations of highly correlated predictors (Spearman’s r > 0.5) were not considered within these two-model sets (Appendix A).

The maximum likelihood was used to estimate coefficients [45]. The relative support of each model was calculated using the Akaike information criterion (AIC) [46]. The models with an AIC score of 0–2 units higher than the lowest observed score had similar and strong support, while models with a score of 4 or more units higher had little support [47]. A model ranking of predictor variables was performed using the R package MuMIn for analysis of an optimal model [48].

Except for when using the AIC for model comparison, the Akaike weight (Wi) and estimate of variation (r^2^) were calculated for each model as a metric of goodness-of-fit [49]. Models were ranked according to their ΔAIC value. To check for spatial autocorrelation in model residuals, we used the ape library (v. 3.0) to calculate Moran’s I coefficients [50]. All statistical analyses were conducted using R, 3.0.1 [51], and significance was assessed at *p* = 0.05.

## 3. Results

From 840 camera-trap days during October 2017–September 2018, a total of 7667 independent capture events of plateau pikas were recorded (9.13 photos per camera trap-day on average). 

### 3.1. Seasonal Activity Patterns

The plateau pikas exhibited unimodal patterns of activity during the cold season from October to April. During the warm season between May and September, the activity pattern of the plateau pikas was bimodal (Figure 3). The activity level of the plateau pikas during the warm season (0.419, CI: 0.401–0.429) was 1.56 times higher than that of the pikas during the cold season (0.269, CI: 0.255–0.272). The analysis showed that 24 h activity varied significantly among months (Appendix A). The activity level of the plateau pikas was highest in August (0.453, CI: 0.408–0.457), followed by March (0.453, CI: 0.408–0.457) and September (0.391, CI: 0.344–0.431), and lowest in February (0.176, CI: 0.162–0.199, Figure 4).

### 3.2. Daily Activity Patterns

The mean daily activity patterns of the plateau pikas varied strikingly among periods (Figure 3). During the cold season, activity rose gradually over the course of the day to a peak near noon followed by a rapid decrease in activity. During the warm season, the activity peaks were in the morning and afternoon. The analysis showed that 24 h activity varied significantly over different time periods during the cold and warm seasons (Appendix A). The activity levels of the plateau pikas during all six time periods of the cold and warm seasons were highest during the daytime (0.268, CI: 0.262–0.277); followed by the time after sunset (0.064, CI: 0.056–0.072), before sunset (0.057, CI: 0.041–0.062), after sunrise (0.055, CI: 0.050–0.059), and before sunrise (0.053, CI: 0.041–0.062); and lowest at nighttime (0.023, CI: 0.012–0.026, Appendix A). According to the statistics regarding activity levels during all six daily periods of the different seasons, the activity of the plateau pikas after sunrise was not significantly higher than that of the pikas before sunset during the cold season (*W* = 0.335, *p* = 0.563, Appendix A), while, during the warm season, the activity of the plateau pikas after sunrise was significantly lower than that of the Pikas before sunset (*W* = 4.352, *p* = 0.037, Appendix A). Activity occurred sporadically during the nighttime but did not show an evident cyclical pattern. The night activity was highest in April.

### 3.3. Effects of Environmental Factors

According to the AIC minimum principle, we found that, for the cold season, there are two GAMMs with a Δ AIC < 4 (Appendix A). Among the fixed factors, TEM and WS were the environmental factors that were significantly related in the optimal model. The model fitting results showed that the activity levels of the plateau pikas decreased with an increase in TEM and WS (Figure 5). For the warm season, there are two GAMMs with a Δ AIC < 4 (Appendix A). Among the fixed factors, TEM and PRE were the environmental factors that were significantly related in the optimal model. The model fitting results showed that the activity levels of the plateau pikas decreased with an increase in TEM and increased at first and then decreased with an increase in PRE (Figure 5). Moran’s I statistic indicated that the residuals had no spatial autocorrelation in the best models of data for either the cold season (*p*-value = 0.61) or the warm season (*p*-value = 0.67).

## 4. Discussion

### 4.1. Seasonal Activity Patterns

Our prediction for the seasonal activity patterns of plateau pikas was supported; we found that the activity of the pikas showed a bimodal pattern during the warm season and a unimodal pattern during the cold season [36]. The bimodal character of plateau pika activity during the warm season reflects their avoidance behavior under heat stress [52]. Our observations of the plateau pika are similar to those of the Royle’s pika (*O. roylei*), which retains a bimodal type of activity from March to October [53]. For small mammals, the influence of light intensity is stronger than that of ambient temperature, which is the main reason for the bimodal pattern [54]. However, maintaining activity at warm times during the cold season may be more critical for pikas inhabiting the QTP, which exhibits long, severe winters and an unpredictable spring snowmelt, leading to the unimodal activity pattern [49].

When comparing the activity levels of the plateau pikas in each month, it was found that the activity levels of the plateau pikas during the warm season were higher than that of the pikas during the cold season; additionally, the top four months in terms of the activity levels of the plateau pikas were August, March, September, and June. We speculate that the high activity levels of plateau pikas in March and June are related to their breeding patterns, and that the high activity levels in August and September are probably related to the storage of food for overwintering. In our study area, the breeding period of plateau pikas is from March to June [55]. Like most lagomorphs, to ensure that energy demands are satisfied during the breeding period, plateau pikas obtain energy by prolonging their time on the ground [56]. By July, the reproductive period of the plateau pikas had already ended as well as the abundance of adequate food resources in the habitat. The plateau pikas needed to be less active to meet their energy demands, so the activity of the plateau pikas was reduced. In September, the temperature drops, and the rangeland enters the withered period. During this period, the plateau pikas began to store food for overwintering [57]. As a result, their levels of activity increased. After October, because of cold weather and snowfall, the plateau pikas began to plug part of their burrows’ openings, causing the tunnels to form microclimatic environments [58]. The temperatures of the burrows were higher than those of the outside environment [59]. The constant-temperature environment in the tunnels reduces the energy consumed during body temperature regulation. This is one of the survival strategies of plateau pikas during the winter, used to minimize their time of exposure to low temperatures and strong winds outside the burrows [60].

### 4.2. Daily Activity Patterns

The results regarding activity levels at different times of the day during different seasons showed that the activity of the plateau pikas peaked near noon during the cold season. During the warm season, the activity peaks were in the morning and afternoon. Our observations are similar to those on the activity of the Japanese pika (*O. hyperborea*), which reportedly increased in the morning and afternoon at low elevations but remained high throughout the day at higher elevations [61,62]. Similarly, the activity levels of other lagomorphs, such as American pikas (*O. princeps*), are reportedly reduced during midday hours at low elevations, but remain high throughout the day at higher elevations [63,64]. However, large-eared pikas (*O. macrotis*) remain active throughout the day [65]. Differences in ambient temperatures during each season accounted for the differences in the daily activity patterns of the plateau pikas. The pika populations generally avoided midday heat during the warm season and daytime high temperatures by restricting high levels of activity to the morning and late afternoon. The results of our best fit GAMM indicated that ambient temperature is the key controlling element in the behavior of plateau pikas. In the alpine meadow inhabited by the plateau pikas, the ambient temperature during the cold season (Mean = 0.7 °C; Range: −16.2~17.6 °C) is much lower than that during the warm season (Mean = 11.8 °C; Range: 2.1~20.2 °C). When stressed by low temperatures during the cold season, the pikas seemed to restrict their daily activity to near noon. In their research on rabbits, Sogliani, et al. [66] found that the activity peak of feral rabbits also appeared at noon during the cold season, which was the result of thermoregulation in feral rabbits. Additionally, a result of their thermoregulation abilities, feral rabbits can also effectively avoid ground predators such as red foxes.

When comparing the activity of the plateau pikas between sunrise and sunset, we found that the plateau pikas were more active before and after sunset than before and after sunrise. These results can be explained by temperature adjustment. During both the cold and warm seasons, the highest temperatures during the day occurred at approximately 16:00, and the lowest temperatures occurred at approximately 7:00 in the morning. To maintain their body temperatures, the plateau pikas attempted to reduce their energy consumption during thermoregulation while foraging by reducing their activity levels under low-temperature conditions. In our study area, the average temperature at sunrise (−0.5 °C) was lower than that at sunset (7.5 °C). Therefore, in addition to activity being unsuitable on very hot days, it is also more suitable before and after sunset than before and after sunrise. We also found that, during the warm season, the activity levels of the plateau pikas after sunrise were not significantly higher than that of the pikas before sunrise or at night, while their activity levels after sunrise during the cold season were considerably higher than they were before sunrise and at night. We believe that this may be related to the foraging of plateau pikas. In the study area, owing to a lack of food resources during the cold season, the plateau pikas were forced to eat during periods of low temperatures to ensure their own energy needs. During the warm season, food resources were more abundant because of appropriate temperatures, allowing plateau pikas to meet their energy needs. Thus, plateau pikas avoid foraging during low-temperature periods.

Our data revealed that plateau pikas are strongly diurnal in the wild and have a very low frequency of nocturnal activity (0.58% of the overall activity), but it is possible for pikas to engage in activities such as burrowing and food store maintenance at night. We also found that night activity was the highest in April. We suggest that higher activity levels at night can be explained by a shortage of food for the animals during their breeding season (from March to June). By that time, plateau pikas need to consume more energy to feed their offspring. The shortage of food resources causes plateau pikas to extend their activity time to night to obtain enough energy. Our study provides evidence for the activity of plateau pikas at night.

### 4.3. Effects of Environmental Factors on the Activity of Plateau Pikas

Small mammals face difficulties performing temperature regulation in cold environments because of their high metabolic rates. As the difference between body temperature and air temperature increases, metabolic rates must increase to compensate for heat loss. As a result, small mammals move their activity to the most favorable periods based on temperature conditions during different seasons [49]. Our results showed that an increase in air temperature had a significant negative effect on plateau pikas. Similarly, the activity of talus-dwelling pikas, such as American pikas (*O. princeps*) and Royle’s pikas (*O. roylei*), is reportedly reduced with an increasing air temperature [47,67]. Royle’s pikas inhabit open, rocky ground and rock talus, which provide a cool and moist refuge during the summer months and insulation from cold in the winter, thereby aiding in thermoregulation [67]. In the alpine meadow where the studied plateau pikas inhabit, the pikas avoided midday high temperatures and took refuge inside their cooler burrows. It is worth noting that the highest daily temperature in our study area occurred at 16:00 each day (Figure 2A), while, during the warm season, the activity peak of the plateau pikas in the afternoon appeared in the period with the highest temperature. In a study on the diurnal variation in solar radiation on the QTP, Liu, et al. [68] also found that solar radiation is the highest at noon. We speculate that the decrease in plateau pika activity at noon during the warm season is mainly due to intense solar radiation rather than the air temperature. We are not aware of any other studies that have found a relationship between the activity of plateau pikas and solar radiation. Whether changes in solar radiation are consistent with changes in the body temperatures of pikas remains to be determined.

In a laboratory study of the actual temperature of plateau pikas, Zong and Xia [36] found that the activity of plateau pikas increased with temperature. At 30 °C, individual death was observed; this is the lethal temperature for plateau pikas. Unlike in the laboratory study, the maximum temperature did not exceed 22 °C throughout our research. The temperature range for plateau pika activity was −10 °C to 10 °C during the cold season and 10 °C to 18 °C during the warm season. There was little activity from the plateau pikas outside these temperature ranges, which differs from what was observed in the laboratory. This is mainly because, under natural conditions, plateau pikas are disturbed by predation risk, food resources, and intraspecies competition, so the tolerance of plateau pikas to high temperatures is reduced [69]. Early studies on the behavior and physiology of American pikas demonstrated that the animals become hyperthermic and die after even brief exposure to moderate ambient temperatures (25.5 °C to 29.4 °C) and intense solar radiation, a combination that prevents thermoregulation [64].

We indicated that precipitation also significantly negatively affected the activity levels of the plateau pikas during the cold season. Precipitation during the cold season mainly restricted the activities of the pikas in two ways. Before the arrival of winter, snowfall in the alpine meadows freezes and thaws over food resources and forms an ice crust, rendering them unavailable [59]. After entering winter, the snow cover provides thermal insulation for pikas exposed to extreme temperatures; adequate snow coverage reduces cold stress and increases survival rates [60]. Therefore, on those particularly snowy days, pikas probably suffered from a serious deficiency of available activity time. The negative effects of precipitation seemed to be less pronounced during the warm season because the rainfall did not impose a thermoregulatory cost on the pikas on hot days. Our results represent a significant step toward understanding the effects of precipitation on the activity of pikas. In our study site, most activity was carried out under conditions of 0–2 mm of precipitation, and the activity levels of the plateau pikas were near zero during precipitation greater than 4 mm.

Research has shown that, on windy days, the activity of small mammals shows a decrease in foraging activities and an increase in time spent in their burrows [70]. In our study, we found similar results in which wind speed during the cold season significantly decreased the activity of the plateau pikas on the daily scale. However, the results of infrared camera monitoring showed that plateau pikas preferred to bask in the sun on the ground or watch at the entrance of their burrows during the warm, windy season, and they chose to avoid the exterior environment outside of their burrows on cold days. This is mainly due to high daytime temperatures on warm days, and the effects of wind in reducing the body temperatures of plateau pikas so that they can resist heat without entering their burrows. On cold days, the ambient temperature is generally low, and the effects of wind accelerate heat loss in plateau pikas. Jackson [71] also found similar phenomena in a study on the factors influencing the rhythm of activity of Brants’ whistling rats (*Parotomys brantsii*). Moreover, wind reduces the activity of small mammals by affecting the transmission of vigilance sounds [64]. An increase in wind speed leads to the rapid attenuation of warning sounds, while the direction of the wind affects the symmetry of signal transmission. This asymmetric sound frequency reduces the stability of the signal, resulting in incomplete and even wrong information [72]. Hayes and Huntly [70] also found that wind speed and direction significantly reduced the effectiveness of alert behavior and restricted the activity level of American pikas (*O. princeps*).

## 5. Conclusions

In conclusion, this study provided evidence for changes in the allocation of activity time for plateau pikas on daily and seasonal scales; pikas’ responses to environmental changes are complex and can vary significantly between the cold and warm seasons. As we demonstrated in this study, plateau pikas occupy a cool and windless microclimate during cold seasons and a cool and humid microclimate during warm seasons to avoid excessive cooling or overheating. These results indicate that changes in habitat microclimatic conditions (specifically, increases in temperature) might significantly restrict the daytime activity of plateau pikas. These results provide baseline information for future research on pikas, especially that which is focused on the behavioral ecology and effects of climate change on the alpine regions of the QTP.

## Figures and Tables

**Figure 1 animals-13-01689-f001:**
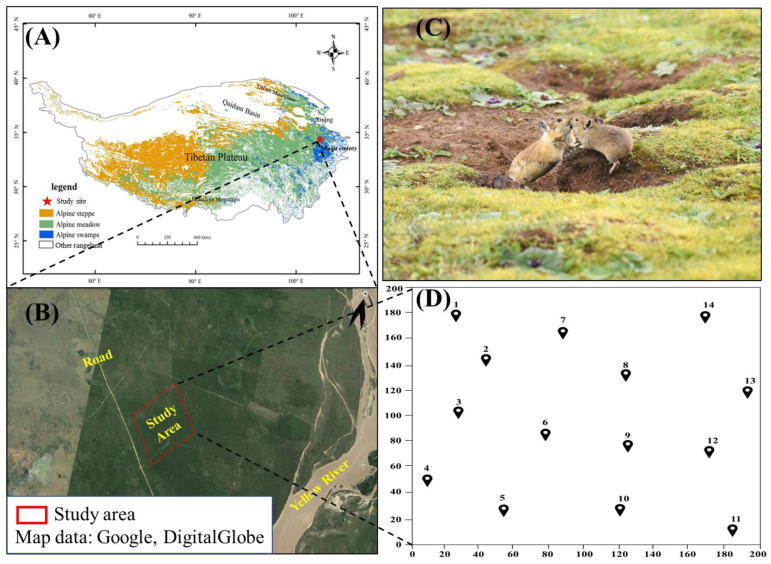
(**A**) Location of the study area on the eastern Qinghai–Tibet Plateau, China. The ArcGIS Pro software (Version: 3.37; URL: https://developers.arcgis.com/ (accessed on 21 May 2019)) was used to draw the map. (**B**) Study site in Maqu County; the map was taken by the author from Google Earth (Version: 7.3.3.7721. URL: https://google-earth.en.softonic.com/). (**C**) Habitat of plateau pikas; photo was taken by Rui Zhou in Maqu County, Gansu Province, China. (**D**) Camera distribution map in the 4 ha (200 × 200 m) plot of Maqu County, Gansu Province (black spots present camera sites, and the numbers 1–14 represents the infrared camera number).

**Figure 2 animals-13-01689-f002:**
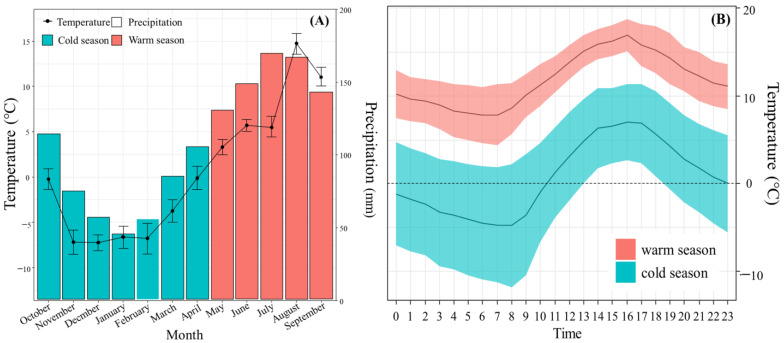
(**A**) Monthly mean ± SD of temperature and total precipitation. (**B**) Mean ambient temperature ± SD as a function of daytime during the cold and the warm seasons.

**Figure 3 animals-13-01689-f003:**
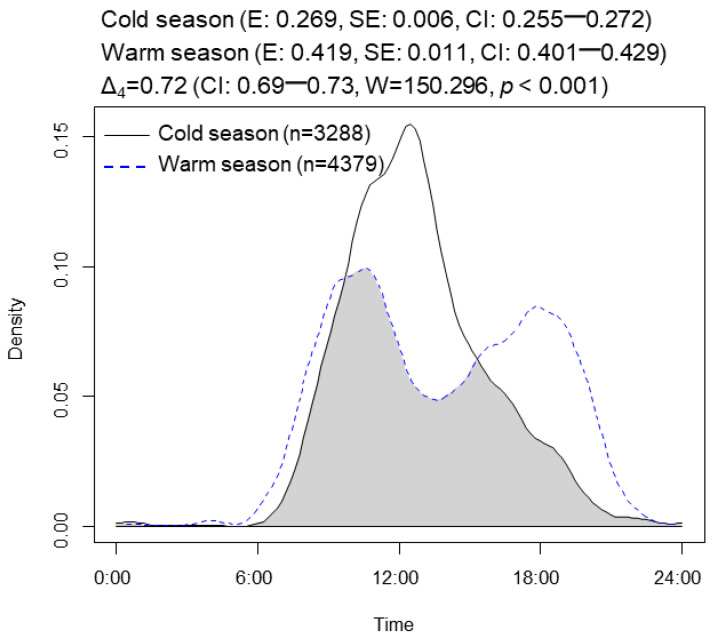
Temporal overlap of plateau pikas between cold and warm seasons. The overlap coefficient is the shaded area. Estimates of activity level (E), standard error (SE), and 95% confidence interval (CI) derived from the fitted distributions of cold and warm seasons are in parentheses. High temporal overlaps are in bold type. *p* (*p*-values): Probability that two sets of circular observations come from the same distribution.

**Figure 4 animals-13-01689-f004:**
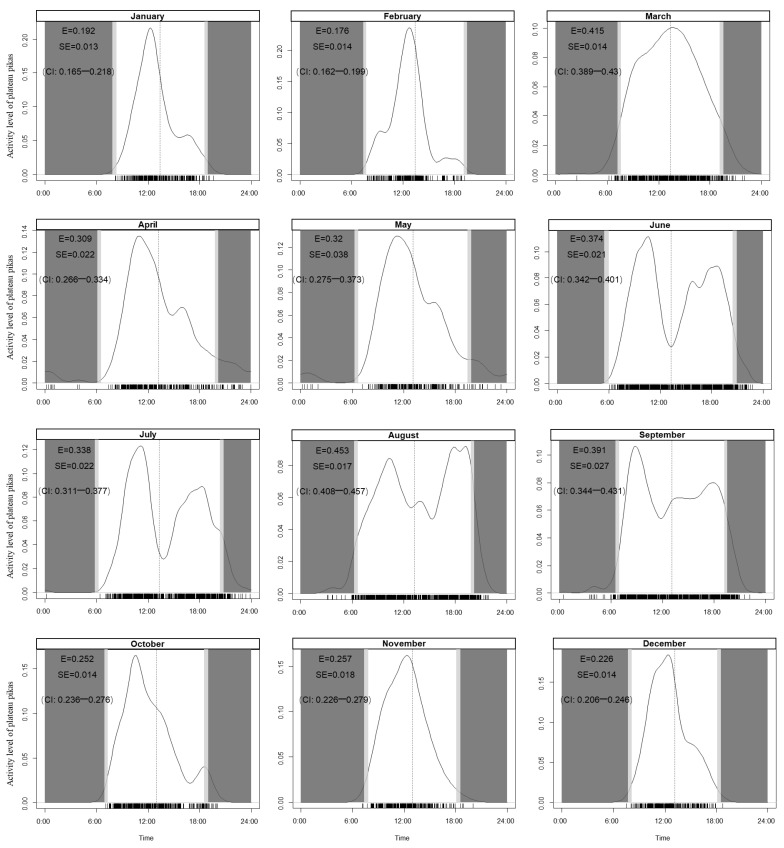
Daily activity patterns of plateau pikas. Y axis and curves are the kernel density estimates of the filming events. Estimates of activity level (E), standard error (SE), and 95% confidence intervals (CI) derived from the fitted distributions are bold; light gray areas represent dawn to sunrise and sunset to darkness, and dark gray areas represent nighttime; short vertical lines above the x-axis indicate the times of individual photographs; note that the widths of these areas vary among seasons as a result of the fluctuating times of sunrise and sunset.

**Figure 5 animals-13-01689-f005:**
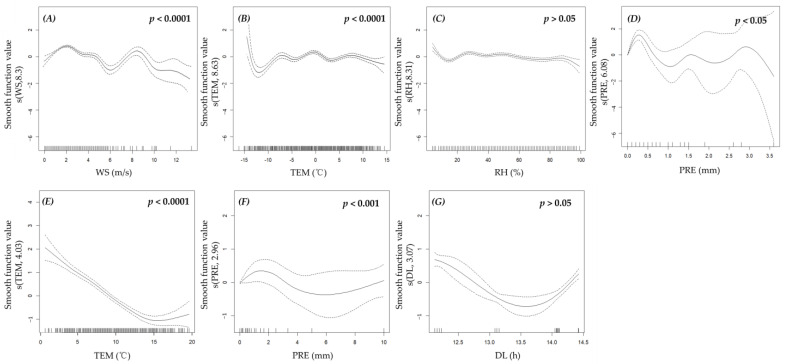
Relationships between environmental factors and activity levels of plateau pikas using optimal GAMM for different seasons. The solid black line represents the curve of the fitting relationship between response and predictor, and the dotted line indicates a 95% confidence interval. (**A**–**D**) represent the environmental factors of the best model for the cold season. (**E**–**G**) represent the environmental factors of the best model for the warm season.

## Data Availability

The data presented in this study are available in Appendix A.

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
