# Peer review of "Daily and Seasonal Activity Patterns of Plateau Pikas (Ochotona curzoniae) on the Qinghai–Tibet Plateau, China, and Their Relationship with Weather Condition"

_animals, 2023, doi:10.3390/ani13101689_

Round 1

Reviewer 1 Report

Dear authors, congrats for the work done. Your research is well-conducted and deserved to be published. Despite this, there are minor problems that must be address regarding lack of references and English style.

In addition to this, I strongly recommend you to analyse your data also with activity and overlap packages to generate 2-3 more figures about activity patterns that must be shown before figure 3.

Here attached the revision file. With 1-2 weeks of additional work you can publish this interesting manuscript.

best,

the reviewer

Reviewer 2 Report

I thank the editor and authors for the opportunity to review this paper. I apologize for my delay. However, I have tried to provide as thorough a review as I am able, and I hope that the authors find my suggestions helpful.

In this paper, the authors use trail cameras in conjunction with daily environmental measurements to assess the diel activity patterns of plateau pika across seasons and environmental conditions. I commend the authors for a job well done. The paper is statistically sound, informative, and impactful. The results are insightful, and I think the readership of Animals will enjoy this work. I do have a few minor recommendations and suggestions, however, that I provide in more detail below, mostly on the Discussion section.

I’ve structured this review by section, with general comments stated first, when applicable, followed by more specific line comments provided afterwards.

One thing to consider: Although well-written, in general, there are multiple grammatical errors throughout that make certain sections a little difficult to understand. Please revise extensively.

Another general comment: Much of the language throughout has a very matter-of-fact tone, with much of these claims having no references that follow. Therefore, many of my line suggestions are simply to add references to these types of claims, and I do so repeatedly throughout the document. When I have a particular reference in mind, I’ll suggest it, but some of them I am too unfamiliar with. Furthermore, this type of language, even with a preponderance of evidence provided, is not recommended. If all of these claims are ‘proven’ facts, then why conduct this investigation at all? I suggest a major reforming of the way the authors frame their discussion, in particular, to highlight that evidence may suggest something, but that much of this information is far from certain.

ABSTRACT:

There is no reference to the field methods used, which will be important for some readers. Specifically, I was impressed with these methods, and I think directly referring to them here will spark interest from similarly-minded readers.

Lines 14-15: I think reference to season is warranted. It seems to be a central tenet of the investigation. It is also talk about in the results below without reference.

INTRODUCTION:

Lines 44-46. I do not understand what it means to be a ‘time minimizer’ or ‘net energy maximizer’. Are these common terms? Please explain what these mean in terms of animal behavior theory.

Lines 49-51: Since much of this work seems to have an extension to climate change, I would also include reference to that here.

Lines 53-55: Although this may be true for small mammals, I do not think it is a universal fact across all taxa, even including larger mammals. I suggest revising to include that, here, the particular interest is in small mammals.

Lines 56-58: Needs a reference here

Lines 59-61: What does, ‘decreasing the behavior of individuals sampled’ mean? Is this in reference to activity?

Lines 61-63: Requires a reference. I suggest: Blount et al., 2021. Biological Conservation 256:108984.

Lines 66-68: Requires a reference. I suggest: Frey et al., 2017. Remote Sensing in Ecology and Conservation 3:123-132.

Lines 71-72: Requires a reference.

Lines 89-91: This sentence to be the first sentence of a new paragraph.

Lines 89-99: Again, there is very little reference to season being a central tenet of this investigation. As it most definitely is, I strongly believe it needs to be directly including the in major postulates and objectives stated here.

MATERIALS & METHODS:

Lines 154-167: First it is stated that independent photographs were those taken more than 30 minutes apart from one another, but then later in the paragraph it is stated that consecutive photographs were considered independent after a 1-hour interval. Which is correct? Please revise.

Lines 168-180: Since data were taken across seasons, were these data transformed into solar time? If not, I highly recommend this transformation, as it may directly affect results when grouping into specific blocks. It looks like some type of transformation was performed, but it is not clear what this is. Speaking from experience, not transforming the data when conducting temporal analysis across seasons can significantly alter results.

RESULTS:

Lines 211-212: Are these the total number of detection events of all species, or only Pika?

Lines 219-221: Looks like September is higher than July. I would switch the order these are presented.

Lines 259-271: Would really like to see the magnitude of these effects written in this paragraph, along with their 95% Confidence Intervals. As written, it is impossible to tell whether these effects were strong, weak, statistically meaningful, etc., and the figure does not clear all of this up.

FIGURE 2: I suggest having the same order of the factors along the y-axis across the two plots, which would allow for an easy visual for readers to compare. Furthermore, there is no key for what the asterisk means in relation to statistical significance. Specifically, what does 2 vs. 3 mean? Z-scores seems like an odd number to include and makes interpretation of the results a little murky.

DISCUSSION:

This section needs the most work, as stated above, much of what is presented, even when obviously alluding to previous research, is done so without proper citation. Furthermore, much of what is presented is presented as absolute factual knowledge, instead of current understanding, which is inaccurate and begs the question why this research was conducted in the first place. The information provided is sound, but where is the preponderance of evidence?

Lines 288-289: Where is the reference?

Lines 299-316: This is definitely not common knowledge. Yet, there is not a single reference in this entire paragraph! Where is all of this information coming from? As presented, it seems to have been pulled out of nowhere. Please cite all sources so readers know where this information can be found.

Lines 394-396: Reference needed.

Lines 396-398: Seems unnecessary.

Lines 424-426: Reference needed.

Lines 434-435: Reference needed.

Lines 437-439: Reference needed.

Reviewer 3 Report

The manuscript submitted by Zhou et al. on the activity patterns of the plateau pika is of interest, but in my opinion, it needs major revisions before it can be accepted for publication in Animals. While some points are missing, the manuscript is often redundant and should be shortened. Even though I am not a native English speaker, I also think that the language must be improved. Finally, statistics should be revised. More details are following.

Specific comments:

Line 77: potyandrous —> polyandrous

Line 105: 3434 m —> 3434 m a.s.l.

Is it to say that the study area is flat?

Lines 106-107 are unclear. I suggest “Minimum and maximum air temperature averaged (±SD) … and … in the cold season (October to April), and … and … in the warm season (May to September)”.

Line 109. According to the graph, minimum temperature seemed to occur around 7:00.

Lines 110-111: “and the daytime… sunrise” can be deleted.

Line 120. AicGIS or ArcGIS?

Lines 121-124. Caption should be shortened.

Fig. 2. The two graphs should be exchanged. They are cited in the reverse order in the main text. A name (e.g. “Daytime hour”) is needed for the x-axis in the graph currently on the left-hand side.

Line 126: “Diurnal… precipitation.” can be deleted.

Lines 126-127. Do the authors mean “(A) Mean ambient temperature ± SD as a function of daytime in the cold and the warm seasons.”?

Lines 127-130: “; X axis… season” should be deleted.

Lines 130-131: “; Bar indicated… temperature” should be deleted.

Lines 133-136. I suggest “Climatic factors were recorded with… The data logger recorded ambient temperature, ambient relative humidity, and wind speed at 1-hr intervals…”

Line 141: I suggest “targeting the entrance(s?) of pikas’ burrows between…”.

Lines 157 and 163: 30 minutes or 1 hour?

Lines 160-162. RAI should be defined more clearly. Here, it seems to be the mean number of independent events per recording day (multiplied by 100). However, in the models compared using AIC, RAI is the number of independent events within a single day (line 184), and in Fig. 3, it seems to be the number of independent events within a single period of the day.

Line 168. To analyse

Lines 168-180. In the present case, the authors did not consider the camera as a random effect factor in contrast to what they did for other analyses. Is it to say that in the present case the variable on which tests were performed is not the value obtained per day and camera, but the mean over the cameras? I am also somewhat surprised that the authors did not transform their data using log(x+1). This would probably have reduced heteroscedasticity. Another possibility was to use generalised linear mixed-effect models (GLMM; family: poisson; link function: log) on the number of independent events, with offset(time) in the fixed part of the models when it was necessary to take into account the time interval.

Lines 181-191. I suggest to delete the first two sentences, and begin the paragraph by “To examine…”. Since the response variable is the “daily RAI”, I suppose that the climatic variables are daily means. This must be indicated.

Lines 190-191. In my opinion, the authors should consider all the possible models.

Table 1 is not useful. It should be deleted or become a supplementary table.

Line 195. fit —> fitted

Lines 196-197. “and compare these nested models (Pinheiro & Bates 2000)” should be deleted. The models that the authors compared are not all nested!

Lines 197-198. Using AIC for comparing numerous models fitted by maximum likelihood is suitable. In contrast, using p-value to discard interaction terms before comparing the remaining models using AIC is a strange procedure. The authors consider a rather high number of predictor variables. Usually, in such a case, no interaction terms are considered in the models, or only the interaction terms that are a priori relevant.

Lines 199-201: “The models… (Wilkening et al. 2015 ).” can be deleted.

Lines 201-202. I don’t see in the Results section where model averaging was used.

Lines 195-209. These paragraphs are especially unclear, and I think that the corresponding analyses should be revised. Using AIC is a suitable procedure for comparing numerous models fitted by maximum likelihood. However, in my opinion, the authors should retain either the model with the best AIC value, or the most parsimonious model with ∆AIC ≤ 2.0, or they should use model averaging (over all the envisaged models, over the models better than the null model, or over the best models giving ∑wi ≈ 0.95). Furthermore, at least some climatic variables such as temperature can be expected to have a non-linear effect (this is also suggested by Fig. 5). As a consequence, I wonder whether each model including temperature should not be duplicated into a version with TEM and another version with TEM+TEM^2.

Line 212. percamera —> per camera

Lines 213-232. These paragraphs are too long and include redundancies. I suggest to reorganise this part of the Results section with a paragraph centred on Fig. 3A then another centred on Fig. 3B.

Fig. 3A. The y-axis has no name. What is the variable corresponding to this axis? Furthermore, was each curve obtained computing a moving average? In this case, this must be indicated in the figure caption or in the Methods section (with the width of the window over which the average is computed). If a more sophisticated method was used, this must be indicated in the Methods section.

Fig. 3C-D. In my opinion, these graphs should be deleted.

Lines 241-271. This section must be rewritten and should focus on the model retained for each season, or alternatively on the average model obtained for each season.

Fig. 4. Such a figure should be replaced by a table. In addition, this table should give the coefficients’ estimated values for the model that has been retained using AIC, or alternatively the average model. (The model involved in the present case may be the average model, but this is not indicated.)

Line 273. endocrinological?

Fig. 5. If this figure is conserved in the revised version of the manuscript, it must be indicated what is “density”, and how it was computed.

Discussion should be substantially shortened. It includes a number of redundancies.

Line 341. According to Fig. 2, minimum temperature seemed to occur around 7:00.

Line 445. endocrinological ?

Lines 446-447. “might be able … behavioural adaptation to” can be deleted. Adaptation implies natural selection for a current role, which must be proved and not simply accepted in every case (see Gould & Vrba 1982 “Exaptation – a missing term in the science of form” Palaeobiology 8(1) 4–15).

Round 2

Reviewer 1 Report

Dear authors, I'm glad you re-worked on your manuscript also adding the analyses I asked you. Now the manuscript has greatly improved its quality.

I just ask you to correct some minor things so I will provide you "minor revisions" (mandatory) that can be assessed in a few minutes.

Here below you see the required changes but i have also attached for you a word file. 

Best,

the reviewer

Revision animals-2272871-peer-review-v2

Line 26: Change in “effects of climate change in some Chinese alpine regions”. Thanks

Lines 34-36: Great, it was exactly what I meant to do!

Line 208: Check style of citation 58. Thanks

Line 210: Missing citations.

Figure 3 à Nice!

Line 268: Missing space.

Line 356: there is a capital “T” in “The”. Put it lower letter. Thanks

Lines 388-392: put a dot (full stop) after “al” à  et al. (2021). Substitue “hares” with “rabbits”. In those lines, put both “rabbit” and “fox” in plural form: “rabbits” and “foxes”. NOTE: you haven’t cited this article in the reference list! à Sogliani D, Cerri J, Turetta R, Crema M, Corsini M, Mori E. (2021). Feral rabbit populations in a peri-urban area: insights about invasion dynamics and potential management strategies. European Journal of Wildlife Research 67:60. Please cite the manuscript. Thanks

Reviewer 3 Report

I thank the authors for partly following my suggestions and recommendations. However, a number of redundancies still lengthen unnecessarily the manuscript; I propose below some changes, but others should be made to reduce the manuscript length. In addition, and this is now the main flaw of the manuscript in my opinion, the analysis of the effects of climatic variables on the plateau pika’s activity level remains unclear, if not inconsistent. To begin with, the authors should clearly state in the Methods section that they used two different procedures: (1) the comparison of linear models, most of which combine the effects of several climatic variables, (2) an analysis considering independently each climatic variable and similar to that used to investigate activity rhythm. The two procedures do not only differ in the number of variables simultaneously taken into account: with procedure (2) a climatic variable can have a non-linear effect, whereas with procedure (1) climatic variables can only have linear effects (in my first report, I proposed to enter ambient temperature in certain models as TEM+TEM2, but the authors did not follow my proposal). Finally, procedure (1) as currently carried out by the authors is unclear and inconsistent. First, the authors envisage a number of models with interaction effects, but these models are not identified in Table S2. Second, whereas the authors found that certain models including interaction effects were among the best models (Table 1 and 2), they only averaged models without interaction terms to obtain Table 3. Instead, they should average the models that they list in Tables 1 and 2, and give the obtained coefficients ± SE in Table 3. Alternatively, the authors could perform procedure (1) only envisaging additive models. In the latter case, however, Tables 1 and 2 would only include models with a single climatic variable. The authors must choose.

Specific comments:

Lines 13-14. I suggest to replace these two sentences by the single following sentence: “Animals are subject to daily and seasonal fluctuations of climatic factors affecting their thermoregulation.”

Line 18. “We showed that” can be deleted.

Line 19. “.”, not “;”.

Lines 124-125. “over a single year” should be deleted.

Line 134. “Minimum and maximum air temperature ranged” can be deleted.

Lines 136-138. Sentences in black should be deleted.

Current version of Fig. 1(D) was perhaps suggested by another reviewer, but I preferred the previous version (picture showing the landscape of the study area). Could the current version of Fig. 1(D) become a supplementary figure S1 ?

Line 153. Though “hm2“ is correct, “ha” is more usual.

Line 159. “(ambient temperature, ambient relative humidity, wind speed)” should be deleted. These parameters are listed in the following sentence.

Lines 160-161. “within the family burrow system territory” —> “within the study site”.

Lines 167-171. I suggest instead: “We installed 14 camera traps (East Red Eagle, Shenzhen Ereagle Technology Co.Ltd, 167 China), each targeting the burrow entrances of a pika family. The cameras were operated for 5 consecutive days every month between October 2017 to September 2018.”

Lines 182-188. I suggest instead: “Pika pictures from the same camera were considered as independent either when they were taken at least 30 min apart, or when they were taken at a shorter interval but unambiguously concerned different individuals.” Please note that the acronym “IPs” did not appear elsewhere in the text.

Line 195. “suncalc package” —> “suncalc R package”

Line 198. “a non-parametric circular kernel-density function”. Evasive… The authors should give more details about the kernel function they used. It probably depends on a parameter controlling the kernel extend.

Line 199. “were” —> “was”

Lines 200-201. “extend” should be deleted, as well as “, taking the minimum of the density functions from two set of samples being compared at each point in time”.

Lines 202-203. I suggest: “This coefficient is the area lying under both of the density curves, and ranges from 0 (no overlap) to 1 (complete overlap) [26, 38]”.

Line 206. “10000” —> “10,000”

Lines 206. “in R environment v.2.6 206 [43]” should be deleted.

Lines 212-213. This sentence should probably be placed at the end of the Data Analyses section.

Line 228. “an information criterion” —> “Akaike information criterion”

Line 246. 95% CI ?

Lines 283-300. Speaking of correlation is not appropriate. Please rewrite this part, for example as follows: “According to the linear models that best fitted the data (Tables 1 and 2), the climatic variables that primarily influenced the activity level of plateau pikas were TEM, PRE, and WS during the cold season, and TEM, PRE, and RH during the warm season. Overall, according to the average model (Table 3), TEM and PRE had a negative effect, whereas…”.

Tables 1 and 2. I suggest to fuse these two tables into a single one. The captions are almost exactly the same.

Lines 303-312. I suggest that this part, which corresponds to Fig. 5 and what I call above “procedure (2)”, becomes another paragraph beginning by “Actually, the activity level of plateau pika increased with the rise of TEM and peaked…”.

Caption of Table 3 (—> Table 2, provided Tables 1 and 2 were fused): I suggest to reduce the caption to “Estimate, adjusted SE, z statistic and P value, obtained by model-averaging across the models of Table 1.”

Table 3. “Dependent” must be deleted.

Round 3

Reviewer 3 Report

I thank the authors for partly following my suggestions, and I appreciate that they improved their statistical analyses. However, in my opinion, some of these analyses can be clarified or improved further, and others might actually be incorrect.

Using generalized additive (mixed) models for analysing the effects of climatic variables on the plateau pika’s activity level is certainly a good idea. However, it seems that in their models the authors used a binary variable (detection / no detection) as response variable, and therefore the binomial family. The number of (independent) detections per day (and site) and family=poisson(“log”) might be more judicious in the present case.

Furthermore, it is unclear whether the authors did or did not perform model averaging on the generalized additive (mixed) models they envisaged. Fig. 5 might actually correspond to the global model (i.e. the model including all the predictor variables). In addition, Table S8 gives a single estimate for each factor whereas several coefficients are generally expected for a GA(M)M. Moreover, Table S8 can hardly be the average of the best models: DL is absent from the best models listed in Table S7 for the cold season.

Finally, if the authors used the ‘overlap’ package to obtain the density curves of Fig. 6, I suspect a problem. I could be wrong here, but I think that the ‘overlap’ package estimates density curves using von Mises kernel. Such a kernel is appropriate for a circular variable such as daytime, but inappropriate for a non-circular variable such as a climatic variable. Be this as it may, generalized additive (mixed) models combining the effects of several climatic variables are better than an analysis considering independently each climatic variable. I therefore suggest to delete the analysis considering independently each climatic variable (and thus Fig. 6).

Specific comments

Line 76. ‘From’ —> ‘from’

Lines 78-79. ‘linear mixed-effects models (LMM)’ —> ‘generalized additive mixed models (GAMM)’

Line 282. ‘Minimum and maximum air temperature’ —> ‘Air temperature’

Line 284. ‘air temperature ranged’ should be deleted

Line 367. ‘of pika family’ —> ‘of a single pika family’.

Lines 368-371 ‘In order to make… 5 consecutive days every month.’ In order to avoid redundancies, these two sentences should be deleted, provided line 367 has been modified as suggested above.

Lines 459-477. For the sake of clarity, I suggest to split this paragraph into two, and write these two paragraphs as follows:

First paragraph: ‘We used the ‘overlap’ R package [41] (and thus the method developed by Ridout and Linkie [26]) to determine the daily activity patterns of plateau pikas in the cold and warm seasons, then quantify the overlap between these two patterns. Accordingly, the 24-h density curve of each season was estimated using a von Mises kernel (default smoothing parameter = 1). Subsequently, the area (Δ) lying under both density curves was assessed using estimator Δ4. This area ranges from 0 (no overlap) to 1 (complete overlap) [26, 38]. Its 95% confidence interval (CI) was further computed using smoothed bootstrap with 10,000 resamples [41].’

Second paragraph: ‘Moreover, we divided the 24-h cycle into six periods: one hour before sunrise, one hour after sunrise, midday, one hour before sunset, one hour after sunset, and midnight [39]. Sunrise and sunset times were obtained from the getSunlightTimes function of ‘suncalc’ R package [40]. Activity level (the proportion of time that pikas spent active) was then estimated for each period using the fitact function of ‘activity’ R package, and compared between periods using the Wald test performed by the compareAct function of the same package [20]. The same method was used to compare the activity level of plateau pikas between months and between seasons.’

Lines 478-479: ‘we conducted two kinds of analysis. First,’ must be deleted if, as I recommend above, the second method is abandoned.

Lines 484-485. I recommend to abandon this method and thus to delete this sentence. If the method were not abandoned, the sentence should become the last sentence of the paragraph.

Lines 486-488 ‘We evaluated the relative support … in the warm season (May-September) (Table S3).’ I suggest instead: ‘Twenty-tree candidate models were envisaged for the cold season (October-April), and 20 for the warm season (May-September) (Table S3).’.

Lines 491-492. ‘Interaction terms were omitted if not significant (p-value > 0.1).’ must be deleted. The authors now use additive models.

Lines 495-497. Did the authors average different generalized additive models they envisaged for a given season, or did they simply consider the global model (especially for obtaining Fig. 5)? Why such a sentence since the authors evoke above 23 candidate models for the cold season and 20 for the warm season?

Line 623. ‘optimal’, ‘average’ or ‘global’? Table S8 gives a single estimate for each climatic variable. It does not seem to correspond to a GA(M)M or an average GA(M)M.

Line 627. See comment for line 623.

Caption of Fig. 5. Does Fig. 5 correspond to an average model or the global model (i.e. the model including all the predictor variables)?

Lines 639-676. This part must be deleted if, as I recommend, the method is abandoned.

Caption of Table S7. There is no “K” in the Table.

Caption of Table S8. Models of Table S8 are not the averaged models of Table S7.

Round 4

Reviewer 3 Report

I thank the authors for following my suggestions. I am pleased to say that in my opinion their manuscript can now be published in Animals. I suggest below (only) three minor modifications.

Lines 195-197. I suggest: “… to conduct generalized additive mixed models (GAMM)[43], family = poisson, link function = log. We used the number of independent detections per day for each camera as a response variable, and used air temperature (TEM), …”.

Line 203: “(Table S1,S2)” —> “(Tables S1, S2)”

Line 262. “(Table S8)” should be deleted.

Best regards